# Online MAP Inference of Determinantal Point Processes

**Aditya Bhaskara**
School of Computing
University of Utah
bhaskaraaditya@gmail.com

**Amin Karbasi**
School of Engineering & Applied Science
Yale University
amin.karbasi@yale.edu

**Silvio Lattanzi**
Google Research Zürich
silviol@google.com

**Morteza Zadimoghaddam**
Google Research Cambridge
zadim@google.com

## Abstract

In this paper, we provide an efficient approximation algorithm for finding the most likelihood configuration (MAP) of size $k$ for Determinantal Point Processes (DPP) in the online setting where the data points arrive in an arbitrary order and the algorithm cannot discard the selected elements from its local memory. Given a tolerance additive error $\eta$, our ONLINE-DPP algorithm achieves a $k^{O(k)}$ multiplicative approximation guarantee with an additive error $\eta$, using a memory footprint independent of the size of the data stream. We note that the exponential dependence on $k$ in the approximation factor is unavoidable even in the offline setting. Our result readily implies a streaming algorithm with an improved memory bound compared to existing results.

## 1 Introduction

Probabilistic modeling of data, along with complex inference techniques, have become an important ingredient of the modern machine learning toolbox. To introduce structures in such models, such as diversity, sparsity, or non-iid samples, while ensuring computational tractability we typically need to provide fast sampling and computationally efficient inference. Determinantal point processes (DPP) are elegant probabilistic models of repulsion that admit such criteria, namely, efficient sampling, marginalization, and conditioning [Kulesza and Taskar, 2012a,b]. They have first been introduced by [Macchi, 1975] in quantum physics to model negative interactions among particles. In recent years, DPPs have found numerous applications in machine learning that rely on a diverse subset selection, such as different forms of data summarization [Mirzasoleiman et al., 2013, Kulesza and Taskar, 2012b, Feldman et al., 2018, Gong et al., 2014], multi-label classification [Xie et al., 2017], recommender systems [Lee et al., 2017, Qin and Zhu, 2013], to name a few.

To define a DPP more precisely, let us assume that we have a set of $n$ vectors $\mathcal{V} = \{v_1, v_2 \ldots, v_n\}$, each of dimension $d$. Define $V = [v_1; v_2; \ldots; v_n]$ to be a matrix of size $n \times d$ (row $i$ consists of vector $v_i$) and construct a positive semi-definite kernel $\mathbb{L} = VV^T$. The entry $(i,j)$ of the kernel $\mathbb{L}$ encodes the similarity between two vectors $v_i$ and $v_j$ as the inner product between them, namely, $\mathbb{L}_{i,j} = \langle v_i, v_j \rangle$. The essential characteristic of a DPP is that the inclusion of an item makes the inclusion of similar items less likely. More specifically, a DPP assigns the probability of sampling a set of vectors indexed by $S \subseteq \mathcal{V}$ as follows:

$$\Pr(S) \propto \det(V_S V_S^T) = \text{vol}^2(S), \tag{1}$$

where $V_S$ is a submatrix of $V$ whose rows consist of vectors in $S$ and $\text{vol}^2(S)$ denotes the volume of the parallelepiped formed by vectors in $S$. Note that when $v_i$ and $v_j$ are similar (under the inner product similarity measure), their vectors are relatively non-orthogonal, and therefore, sets that include both of them have less volume. This way, DPPs incentivizes diversity and encodes negative correlations among elements. Note that the normalization factor in (1) can be computed explicitly and is equal to $\det(\mathbb{L} + \mathbb{I})$.

A fundamental problem associated to any probabilistic model, and in particular a DPP, is to find the most likely, or the maximum a posteriori (MAP), configuration [Gillenwater et al., 2012]:

$$\arg \max_{S \in \mathcal{I}} \Pr(S) = \text{vol}^2(S), \tag{2}$$

where $\mathcal{I}$ indicates some feasibility constraints, most commonly a size constraint $|S| \leq k$. We denote the optimum value of problem (2) by OPT. It is known that finding a solution to (2) is NP hard, even if we tolerate an approximation factor of $c^k$ for some $c > 1$ [Summa et al., 2015]. Using elegant optimization techniques, the works of [Nikolov, 2015, Nikolov and Singh, 2016, Ebrahimi et al., 2017] and others have developed algorithms that have an approximation ratio close to $e^k$, nearly matching the lower bounds. However, all of these algorithms require the entire dataset to be in memory, and have impractical running times.

Motivated by large-scale applications, recent work [Indyk et al., 2018, Mahabadi et al., 2019] has studied algorithms in sublinear models such as data streaming. Here, data points arrive one after another and the algorithm needs to be able to maintain a near optimal solution, while (a) using only a sublinear amount of memory and (b) having small per-point processing time. Very recently, using previous work on coresets, [Mahabadi et al., 2020] showed that for any $\epsilon > 0$, there exists a streaming algorithm that uses memory $O(\frac{1}{\epsilon}n^\epsilon kd)$, while achieving an approximation factor of $\tilde{O}(k)^{(k/\epsilon)}$ for the objective (2). This yields a trade-off between the space complexity and approximation factor.

The focus of our work is the *online* model, which is even more restrictive than streaming. As in streaming, our goal is to process points quickly as they arrive while maintaining small space. Additionally, we have the restriction that once a point is added to memory, it remains in memory for the remainder of the algorithm. The online model is preferred in applications where we need to maintain a solution at any point in time and where we want the solution to be consistent and *stable*, i.e., changes in the solution occurs rarely and only when "really necessary" [Cohen-Addad et al., 2019, Jaghargh et al., 2019, Lattanzi and Vassilvitskii, 2017, Bhaskara et al., 2019]. Our goal is to develop algorithms whose space and time complexity per vector are independent of $n$, while achieving an approximation ratio competitive with the best known streaming algorithms. In general, this is impossible: consider a simple example where we receive groups of $k$ vectors, with each group having orthogonal vectors of a geometrically increasing length (factor $\beta$). Since we do not know the length of the stream, any algorithm with an approximation factor better than $\beta^k$ must select all the points.[1] Any competitive algorithm must thus make some (possibly mild) assumptions on the full dataset. Our contributions are as follows:

- A simple online variant of local search (denoted ONLINE-LS). We prove that this algorithm achieves an approximation ratio $k^{O(k)}$, matching the best known streaming algorithms. However, the space complexity depends (logarithmically) on $\Delta$, an appropriately defined condition number parameter. (See Theorem 3.1.)

- (Main contribution) An algorithm ONLINE-DPP that avoids conditional number dependence, but incurs a small additive error. Given a parameter $\eta$, the algorithm keeps only $\text{poly}(k, \log(1/\eta))$ vectors, while incurring a multiplicative approximation factor of $(k + \log(1/\eta))^{O(k)}$ along with an additive error $\eta$. (See Theorem 3.2.)

- Experiments demonstrating the efficiency of online methods for MAP inference. We demonstrate how even the simple algorithm ONLINE-LS finds solutions that compete favorably with *offline* algorithms (that store the entire dataset in memory).

We also bound the amount of *recourse* in our algorithms, i.e., the number of times the solution changes through the course of the stream. This number will be independent of $n$ in both algorithms.

## 1.1 Related work

The problem of finding the MAP configuration of a DPP has been studied extensively, and has also been referred to by other names such as sub-determinant maximization and D-optimal experiment design. Much of the work can be divided into a few strands, which we discuss briefly.

**Submodular Maximization.** It is known that the set function $f(S) = \log \det(V_S V_S^T)$ is a submodular function. Therefore, a series of previous work applied submodular maximization algorithms such as greedy addition [Chen et al., 2018, Badanidiyuru et al., 2014, Kulesza and Taskar, 2012b], soft-max relaxation [Gillenwater et al., 2012], multi-linear extension [Hassani et al., 2019], in order to find the MAP configuration. Interesting results are also known for maximizing determinantal functions with approximate submodularity Bian et al. [2017] and maximizing submodular functions whose value can be negative Harshaw et al. [2019]. However, there are two drawbacks with such methods. First, even though $f(S)$ is submodular, it might be negative, i.e., when $\det(V_S V_S^T) < 1$. Almost all submodular maximization algorithms assume non-negativity of the objective function in order to provide a constant factor approximation guarantee [Buchbinder and Feldman, 2017]. Second, any $\alpha$-approximation guarantee for a non-negative and non-monotone submodular function $f$ may only provide a $\text{OPT}^{1-\alpha}$ approximation guarantee for problem (2) where OPT is the volume of the optimum solution. Such approximation guarantees may be much worse than multiplicative guarantees.

**Coresets.** Given a large data set, a coreset is defined to be a subset with the property that solving an optimization problem on the subset yields a good solution to the problem on the full dataset Agarwal et al. [2005]. It is a powerful paradigm in the design of sublinear algorithms [Indyk et al., 2014, Mirrokni and Zadimoghaddam, 2015]; for problem (2), Indyk et al. [2018] showed the construction of coresets of size $O(k \log k)$ that yield an approximation guarantee of $\tilde{O}(k)^k$. A more practical construction was proposed in Mahabadi et al. [2019], using simple greedy and local search coreset algorithms, with slightly weaker guarantees. Standard application of such coresets in the online setting implies $k^{O(k)}$ approximation guarantee at the cost of having the space complexity dependent on $\sqrt{n}$. Using similar ideas, [Mahabadi et al., 2020] very recently gave a one-pass streaming algorithm for (2) with a trade-off between the space complexity and approximation as discussed earlier. Our results yield bounds independent of $n$, even in the stronger online model.

In parallel to the MAP inference, there has been a surge of interest in fast and efficient sampling from submodular point processes [Rebeschini and Karbasi, 2015, Gotovos et al., 2015]. In particular, spectral algorithms have been recently developed for generating samples from DPPs [Derezinski et al., 2019] and more generally from strongly Rayleigh measures [Anari et al., 2016]. Such techniques have also been generalized to other negative dependence probabilistic models and more complex constraints [Li et al., 2015, Celis et al., 2016, Mariet et al., 2018].

## 2 Preliminaries

We consider a *stream* of $d$-dimensional vectors $\mathcal{V}$ whose elements $v_1, v_2, \ldots, v_n$ arrive in order. Unless otherwise specified, the algorithm is not aware of $n$, the size of the data stream. Throughout the paper, $k$ will always denote the number of vectors we wish to output as the solution. Following Mahabadi et al. [2019], we sometimes refer to a set of vectors in $\mathbb{R}^d$ as a *point set* and we call the elements either points or vectors.

In this paper, we are interested in solving problem(2) subject to a $k$ cardinality constraint. Let $S \subset \mathbb{R}^d$ be a set of vectors of size $k$. For any set of points $P$, we use $\text{OPT}_k(P)$ to denote

$$\text{OPT}_k(P) = \max_{S \subseteq P, |S| = k} \det(V_S V_S^T) = (\text{vol}(S))^2,$$

where as before $V_S$ is a $k \times d$ matrix whose rows are vectors of $S$. When $k$ is clear from the context, we sometimes drop the subscript and write $\text{OPT}(P)$.

For a set of vectors $S$, $\text{span}(S)$ denotes the linear subspace spanned by $S$. We denote the set of all $k$-dimensional linear subspaces of $\mathbb{R}^d$ by $\mathcal{H}_k$. Finally for a point $p$ and a subspace $\mathcal{H}$, we use $d(p, \mathcal{H})$ to denote the (closest) Euclidean distance of $p$ from $\mathcal{H}$. For a set of points $P$, the *directional height* (as defined in [Mahabadi et al., 2019]) of $P$ with respect to $\mathcal{H}$ is defined as $\max_{p \in P} d(p, \mathcal{H})$.

A core-set for the $k$-directional height is defined as follows.

**Definition 2.1.** *Mahabadi et al. [2019](Core-set for $k$-directional height) Given a point set $P$, a subset $C \subseteq P$ is called an $\alpha$-approximate core-set for $k$-directional height if for all $\mathcal{H} \in \mathcal{H}_{k-1}$,*

$$\max_{p \in C} d(C, \mathcal{H}) \geq \frac{1}{\alpha} \cdot \max_{p \in P} d(P, \mathcal{H}).$$

*We denote an $\alpha$-approximate coreset of a set $P$ by $c(P)$ (clearly, $P$ can have multiple core-sets).*

## 3 Online volume maximization

We now present our algorithmic results for online DPP. The first algorithm (Section 3.1) is very simple to implement, but suffers a dependence on an appropriately defined condition number. We then (Section 3.2) give an algorithm that avoids this dependence, but has additive error in its guarantee.

### 3.1 Online addition with a stash

We use the following notation for stating the result. Let $\text{vol}_{\text{first}}(\mathcal{V})$ be the volume of the first parallellepiped with non zero volume that can be formed in the stream (e.g., if the first $k$ vectors in the stream are linearly independent, then $\text{vol}_{\text{first}} = \text{vol}(v_1, v_2, \ldots, v_k)$). We give a simple algorithm and prove the following.

**Theorem 3.1.** *Let $\mathcal{V} = \{v_1, v_2, \ldots\}$ be a stream of vectors that arrive in an online model. Algorithm 1 (ONLINE-LS) provides the following guarantees:*

1. *At any point, it returns a solution with $k^{O(k)}$ approximation to the $k$-volume maximization problem so far.*

2. *At any point in time, the algorithm keeps in memory at most $O(k + \log \Delta)$ vectors where $\Delta$ is the ratio $\text{OPT}_k(\mathcal{V})/\text{vol}_{\text{first}}(\mathcal{V})$. Also, the solution is selected within those vectors. Furthermore, vectors are never deleted from the algorithm memory.*

3. *The running time of each step is $O((k + k \log^2(\Delta))T_{\text{vol}}(k))$ where $T_{\text{vol}}(k)$ is the time it takes to compute the volume of a size $k$ set of columns.[2] The amortized running time of each step is $O((k + \frac{k \log^2(\Delta)}{n})T_{\text{vol}}(k))$ where $n$ is the total number of columns in the whole stream. For $n \geq \log^2 \Delta$, the amortized computation in each step is only $O(k)$ volume computations.*

4. *The total amount of recourse (number of changes to the solution during the entire stream) is $O(\log \Delta)$.*

The dependence on $\Delta$ is undesirable because $\text{vol}_{\text{first}}$ can be arbitrarily small (all it takes is one set of badly conditioned vectors). We show how to overcome this issue in Section 3.2.

**Outline.** Algorithm 1 maintains a candidate solution $S$. Upon seeing a new column $v_t$, if we can replace some $v_i \in S$ with $v_t$ and increase $\text{vol}(S)$ by a factor $(1 + \epsilon)$, we perform the swap. But instead of throwing $v_i$ away, we store into a "stash" $T$. Every time we perform a swap and update solution $S$, we try to use the columns in the stash to improve solution $S$. Formally, we see if swapping any column $v_\ell \in S$ with $v_m \in T$ improves the volume of $S$ by a factor of $1 + \epsilon$. If such a pair exists, we perform the swap (and move $v_\ell$ to the stash). We stop when no such swap is possible.

**Why a stash?** It is natural to ask why we need to keep the removed vectors in a stash; can they later become useful to the optimal solution? It turns out that this is the case. In the supplement, Section E, we show a simple example with $k = 2$ where no bounded approximation factor is possible if vectors are removed permanently. Very roughly, the vectors in this example $v_1, v_2, \ldots, v_r$ make slowly increasing angles with the $x$ axis, and the lengths increase slightly in every step. This makes LocalSearch (without a stash) always maintain the two most recent vectors as the solution. In the end, the optimal solution turns out to be $\{v_1, v_r\}$ as they have the largest angle. Making this formal requires some more work and is deferred to Section E.

Our proof of Theorem 3.1 combines ideas from [Mahabadi et al., 2019] with properties of our stash based algorithm. The details are deferred to the supplement, Section A. The approximation bound

**Algorithm 1** Local Search with Stash (ONLINE-LS)

---
1: Input: A sequence of arriving columns $v_t$, and a constant $\epsilon$ (set to 1 for simplicity)
2: Output: A subset $S$ of $k$ columns at any time step
3: $S, T \leftarrow \emptyset$
4: **for** newly arrived column $v_t$ **do**
5:    **if** $|S| < k$ and $v_t \notin \text{span}(S)$ **then**
6:       Add $v_t$ to $S$
7:    **else**
8:       **if** $\exists v_i \in S : \text{vol}(S \cup \{v_t\} \setminus \{v_i\}) > (1 + \epsilon)\text{vol}(S)$ **then**
9:          $S \leftarrow S \cup \{v_t\} \setminus \{v_i\}$
10:         Add $v_i$ to $T$
11:         **while** $\exists v_\ell \in S, v_m \in T : \text{vol}(S \cup \{v_m\} \setminus \{v_\ell\}) > (1 + \epsilon)\text{vol}(S)$ **do**
12:            $S \leftarrow S \cup \{v_m\} \setminus \{v_\ell\}$
13:            Add $v_\ell$ to $T$
14: Return $S$

---

of $k^{\Omega(k)}$ turns out to be optimal for the algorithm. We thank the anonymous reviewer for providing the following example: suppose the stream consists of $e_1, e_2, \ldots, e_k$ (the standard basis vectors), followed by $h_1, h_2, \ldots, h_k$, where $h_i$ are the vectors of the Hadamard basis (scaled so that the entries are $\pm 1$), then the algorithm will not add any of the $h_i$ to the solution (because replacing one of the $e_i$ with some $h_j$ does not help). But choosing latter $k$ vectors gives a volume of $(\sqrt{k})^k$, as opposed to 1 achieved by the algorithm.

### 3.2 Additive error algorithm

**Normalization.** For simplicity of exposition, we assume below that all the columns in $\mathcal{V}$ have $\|v_i\| \leq 1$. This assumption can be removed at the expense of replacing $\log(1/\eta)$ in our space and approximation bounds (in Theorem 3.2) with $\log(\max_i \|v_i\| /\eta)$. The algorithm is required to know an upper bound on $\log(\max_i \|v_i\| /\eta)$. Note that since it is a logarithmic term, even a very coarse approximation to $\max_i \|v_i\|$ suffices. Details of this can be found in the supplement, Section C.

The main result is the following:

**Theorem 3.2.** *Given a parameter $\eta > 0$, algorithm 2 (ONLINE-DPP) has the following guarantees:*

1. *At any point, the algorithm returns a solution $\mathcal{S}$ that satisfies*

$$(k + \log(1/\eta))^{O(k)} vol(\mathcal{S}) + \eta \geq \text{OPT}(\mathcal{V}).$$

2. *The space complexity is $O\left(k^3 \log^2 k + k \log^2 \frac{1}{\eta}\right)$ vectors.*

3. *The running time of each step is $O(L \cdot T_{ridge}(kL) + k^3 \log k \cdot L^2 T_{vol}(k))$. Parameter $L = O(\log(1/\eta) + k \log k)$ is defined as $\log(1/\gamma)$ (using (3)). Function $T_{ridge}(kL)$ is the time to compute the ridge leverage score when we currently have $kL$ columns. Function $T_{vol}(k)$ is the time it takes to compute the volume of a parallelepiped formed with $k$ vectors.*

4. *The total amount of recourse (number of changes to the solution $\mathcal{S}$ during the entire stream) is $O(\log(1/\eta) + k \log k + k \log \log(1/\eta))$.*

**Remark.** The space complexity is measured in terms of the number of vectors. By definition, this is upper bounded by $d$, but it can be much smaller if the columns are sparse. Further, as noted in Mahabadi et al. [2020], volume maximization can also be performed after applying a random projection in certain parameter regimes. Also, we note that $T_{ridge}$ has been extensively studied and can be computed in time nearly linear in the number of non-zeroes in the vectors in $S_\delta$ (see [Cohen et al., 2017, Alaoui and Mahoney, 2015], and references therein).[3]

The following definition extends Definition 2.1 to incorporate an additive term.

**Definition 3.3.** *For a set $V$ of vectors, we say that $S$ is an $(\alpha, \gamma)$-core-set for $k$-directional height, if for every subspace $\mathcal{H}$ of $\mathbb{R}^d$ with dimension $(k-1)$, we have:*

$$\max_{u \in S} d(u, \mathcal{H}) \geq \frac{1}{\alpha} \cdot \max_{u \in \mathcal{V}} d(u, \mathcal{H}), \text{ whenever } \max_{u \in \mathcal{V}} d(u, \mathcal{H}) \geq \gamma.$$

Thus the definition does not yield any guarantee when the maximum distance to a subspace is $< \gamma$. We also define a parameter $\gamma$ and a set $\mathcal{D}$ that we use throughout.

$$\gamma = \frac{\eta^2}{(64k)^{2k}} \; ; \; \mathcal{D} = \{2^i \gamma : i = -1, 0, 1, \ldots, \text{ and } 2^i \gamma \leq \frac{1}{2}\}. \tag{3}$$

---

**Algorithm 2** Online Coreset for Additive Error Approximation (ONLINE-DPP)

---

1: **Input:** An additive error parameter $\eta$, and a sequence of vectors $\mathcal{V} = v_1, v_2, \ldots$
2: **Output:** Core-set $S$
3: Define $\gamma$ using (3)
4: For all $\delta \in \mathcal{D}$, set $S_\delta = \emptyset$ and $M_\delta = \delta^2 \mathbf{I}_d$
5: **for** newly arrived vector $v_t$ **do**
6:     **for** each $\delta \in \mathcal{D}$ **do**
7:         **if** $\left(v_t^T (M_\delta M_\delta^T)^{-1} v_t > 2 \text{ and } |S_\delta| < 4k \log(4/\delta)\right)$ **then**
8:             Add $v_t$ to $S_\delta$
9:             Update $M_\delta \leftarrow M_\delta + v_t v_t^T$
10: Return $S = \cup_{\delta \in \mathcal{D}} S_\delta$

---

**Algorithm outline.** The key objects in our algorithm are the sets $S_\delta$, one for every $\delta \in \mathcal{D}$, and the corresponding matrices $M_\delta$. These sets are all initialized to be empty. Once a $v_t$ arrives, for every $\delta$, it checks if the following condition holds: $v_t^T (M_\delta M_\delta^T)^{-1} v_t > 2$. We refer to this as the "$M_\delta$ condition" (the LHS is known as the ridge leverage score [Cohen et al., 2017]). If so, it adds $v_t$ to $S_\delta$. This is done as long as $|S_\delta|$ is small. Once $|S_\delta| \geq 4k \log(4/\delta)$, no more elements are added to $S_\delta$ (this check is done independently for every $\delta$).

Algorithm 2 shows how to maintain a core-set $S$ which is guaranteed to contain an approximately optimal solution. Since our overall goal is to maintain a near optimal solution at every time step, we will also keep a solution $\mathcal{S} \subset S$. Every time we add an element to $S$, we check to see if $\mathcal{S}$ is to be updated. For this, we can use any of the volume maximization algorithms in the literature. For concreteness, we use the local search algorithm in [Mahabadi et al., 2019]. Every time we add a vector to coreset $S$, we run the local search algorithm and let $S'$ be its solution. If $\text{vol}(S') \geq 2\text{vol}(\mathcal{S})$, we set $\mathcal{S}$ to be $S'$. Based on the approximation guarantee of local search [Mahabadi et al., 2019],

$$\text{OPT}(S) \leq (3k)^{2k} \cdot \text{vol}(\mathcal{S}) = k^{O(k)} \cdot \text{vol}(\mathcal{S}) \tag{4}$$

Outputting $\mathcal{S}$ as the current solution provides all the guarantees of Theorem 3.2 except that it may change in too many time steps. To upper bound the recourse (number of updates to the solution), we output the first $k$ vectors in the stream as the current solution until $\text{vol}(S')$ reaches $\tau = \eta/((3k)^{2k} \cdot \alpha^k)$ where $\alpha$ is from Theorem 3.4. After this point, we output $\mathcal{S}$ as the solution.

**Space complexity.** By limiting the size of each $S_\delta$ to $4k \log(4/\delta)$, we obtain a simple bound on the overall space. The number of choices of $\delta$ is $O(\log(1/\gamma))$, and each $\delta \geq \gamma/2$. Thus the space complexity is at most

$$O(\log(1/\gamma)) \cdot 4k \log(8/\gamma) = O\left(k \log^2(1/\gamma)\right). \tag{5}$$

The main technical result is the following:

**Theorem 3.4.** *The set $S = \cup_{\delta \in \mathcal{D}} S_\delta$ returned by the algorithm is an $(\alpha, \gamma)$-core-set for $k$-directional height, where $\alpha = 8\sqrt{k \log(8/\gamma)}$.*

*Outline of the proof of Theorem 3.2.* The space and time complexity analyses are relatively straightforward, using the bounds on $S_\delta$ along with (5). The tricky part is to show the approximation guarantee. Theorem 3.4 shows that $S = \cup_\delta S_\delta$ is an $(\alpha, \gamma)$-core-set for $k$-directional height. So a

natural approach is to start with the true optimum solution $\{u_1, u_2, \ldots, u_k\}$ and show iteratively that there exist "replacements" in $S$. This is easy if we had coresets where the additive error $\gamma$ (see Definition 3.3) is zero. When $\gamma \neq 0$, we need to show that the directional height in each step is bounded from below. This requires a more careful inductive argument, which crucially uses the choice of the parameter $\gamma$ from (3).

The full details of the proof are deferred to the supplement, Section B.1. □

The main step now is to prove Theorem 3.4. We do this by showing structural properties of the sets $S_\delta$. The following *packing lemma* will play a key component in the argument.

**Lemma 3.5.** *Suppose $v_1, v_2, \ldots, v_m \in \mathbb{R}^q$ are vectors of length $\leq 1$. Let $V_i$ denote the matrix with columns $v_1, v_2, \ldots, v_i$. Suppose that the vectors satisfy*

$$v_{i+1}^T (\delta^2 \mathbf{I}_q + V_i V_i^T)^{-1} v_{i+1} > 1.$$

*Then we have $m \leq 4q \log(2/\delta)$.*

The proof is an application of the matrix determinant lemma and Cauchy Schwarz; it is deferred to the supplement (Section B.1.1). The next lemma gives a clean interpretation of the $M_\delta$ condition.

**Lemma 3.6.** *Let $U$ be a matrix with columns being $u_1, u_2, \ldots, u_m \in \mathbb{R}^d$, and let $\delta > 0$. For any $v \in \mathbb{R}^d$, we have the following:*

1. *For any $c > 0$, if $v^T (\delta^2 \mathbf{I}_d + UU^T)^{-1} v \leq c$, then we can write $v = Ux + \delta z$, where $\|x\|^2 + \|z\|^2 \leq c$.*

2. *Suppose $v^T (\delta^2 \mathbf{I}_d + UU^T)^{-1} v > 2$. Then we cannot express $v = Ux + z'$, for any choice of $x, z'$ that satisfy $\|x\| \leq 1$ and $\|z'\| \leq \delta$.*

Once again, we defer the proof to the supplement (Section B.2). We next outline our main result.

*Outline of proof of Theorem 3.4.* We need to show that for any $v_i \in \mathcal{V}$ and any $(k-1)$-dimensional subspace $\mathcal{H}$ of $\mathbb{R}^d$, if $d(v_i, \mathcal{H}) \geq \gamma$, then there exists a $v \in S$ such that $d(v, \mathcal{H}) \geq (1/\alpha) \cdot d(v_i, \mathcal{H})$. Let us fix any such $v_i$ and subspace $\mathcal{H}$. We show that in fact, some element of $S_\delta$ satisfies this property, where $\delta$ is the unique value in $\mathcal{D}$ satisfying $2\delta \leq d(v_i, \mathcal{H}) < 4\delta$. Note that by our choice of parameters, there is always exactly one such $\delta \in \mathcal{D}$. In what follows, let us fix this value of $\delta$, and let $\alpha = 8\sqrt{k \log(8/\gamma)}$ be the core-set approximation factor we are aiming to prove in Theorem 3.4. Since all values in $\mathcal{D}$ including $\delta$ are at least $\gamma/2$, we also have $\log(4/\delta) \leq \log(8/\gamma)$. So it suffices to prove that there exists a $u \in S_\delta$ such that

$$d(u, \mathcal{H})^2 \geq \frac{\delta^2}{32k \log(4/\delta)}.$$

Now, consider the set $S_\delta$ at the $i$th time step of the algorithm (when we see $v_i$). If $v_i$ gets added to $S_\delta$, there is nothing to prove. Otherwise, either the $M_\delta$ condition was not met, or the size of $S_\delta$ is already at the threshold of $4k \log(4/\delta)$. Let us consider the two cases separately and show the following.

**Claim 1.** *For $\delta$ as above, suppose $|S_\delta| < 4k \log(4/\delta)$ at the time that $v_i$ arrives, and suppose that $v_i^T (M_\delta M_\delta^T)^{-1} v_i \leq 2$. Then there exists a $u \in S_\delta$ such that*

$$d(u, \mathcal{H})^2 \geq \frac{\delta^2}{4k \log(4/\delta)}.$$

**Claim 2.** *For some $\delta \in (0, 1/2)$, suppose $|S_\delta| = 4k \log(4/\delta)$. Then for* any *$(k-1)$ dimensional subspace $\mathcal{H}$, there exists $u \in S_\delta$ such that $d(u, \mathcal{H})^2 > \frac{\delta^2}{8|S_\delta|}$.*

Claim 2 is interesting as it shows that some point in $S_\delta$ has a high enough distance to *every* $(k-1)$-dimensional subspace. It is the main step where we use the packing lemma stated above. The full proofs of the claims are in the supplement, Section B.3. These imply that if $|S_\delta|$ was $4k \log(4/\delta)$, it does not matter that $u_i$ was not added to $S_\delta$. For any $\mathcal{H}$, we already have $\max_{u \in S_\delta} d(u, \mathcal{H})$ being $\geq (1/\alpha) \cdot 4\delta \geq (1/\alpha) \cdot d(u_i, \mathcal{H})$. This completes the proof of the theorem. □

# 4 Experiments

In this section we compare the experimental performances of our Algorithm 1 (ONLINE-LS), with:
– an online greedy algorithm, ONLINEGREEDY that upon processing a new row adds it to the solution if, by swapping it with any row in the solution, it is possible to increase the volume.
– the classic offline greedy algorithm, GREEDY that does $k$ passes on the entire dataset and in every pass it adds to the current solution the row that increases the volume the most.

All our experiments have been carried out on a standard desktop computer and all the experiments presented in this section are fully deterministic. In our experiments we consider three standard datasets: the Spambase dataset [Dua and Graff, 2017], the Statlog(or Shuttle) dataset [Dua and Graff, 2017] and the Pen-Based Recognition dataset [Dua and Graff, 2017]. All the datasets contain only integer and real values, the Spambase dataset contains 4601 instances of 57 dimensions, the Statlog dataset contains 58000 instances of 9 dimensions and the Pen-Based Recognition dataset contains 10992 of 16 dimensions. In all our experiments we normalized the norm of the instances to be smaller than 1.

**Quality, consistency and amount of computation.** In Figure 1 we show a comparison of the three algorithms on the Spambase dataset and the Statlog dataset for $k = 8$ and $\epsilon = 0.1$.[4] For the offline GREEDY algorithm we report only the quality of the solution because the running times are incomparable. For ONLINELS and ONLINEGREEDY we report the number of volume computations as system independent proxy of the running time and the number of swap in the solution during the execution of the algorithm to capture the consistency of the algorithm. In all plot for ONLINELS and ONLINEGREEDY we represent the evolution of quality of solution, number of volume computations and number of swaps as a function of the number of columns analyzed so far. For GREEDY we show the final quality of the solution. Interestingly, both ONLINELS and ONLINEGREEDY recover a solution that has quality comparable with the solution of the offline algorithm. Furthermore, both algorithms execute a similar number of volume computations but ONLINELS execute significantly less swaps and so it is more consistent than ONLINEGREEDY.

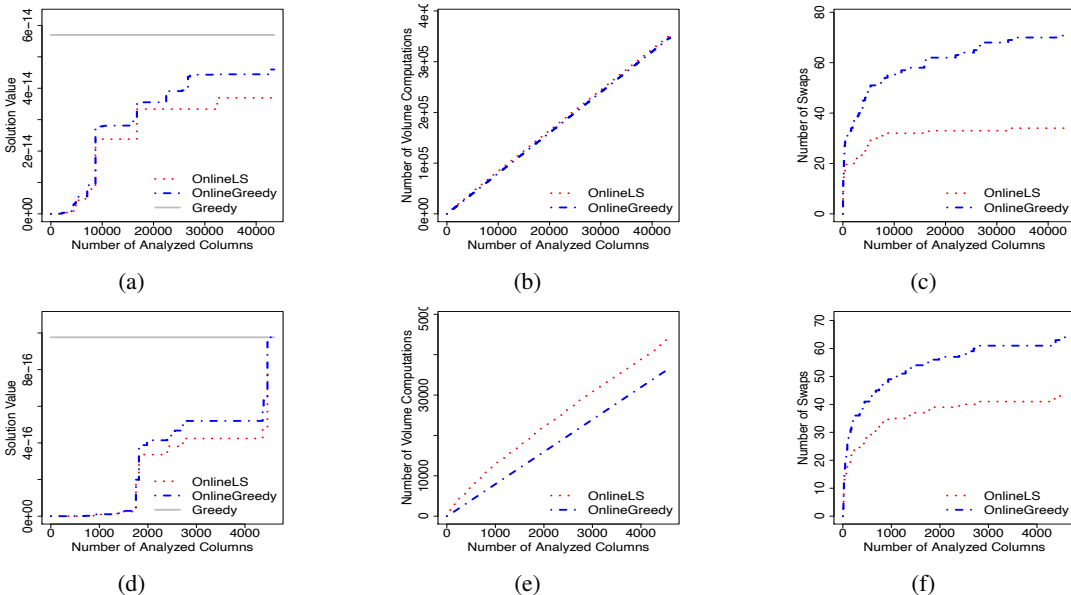

Figure 1: Performances of the algorithms on the Shuttle ((a),(b),(c)) and Spambase datasets ((d),(e),(f)) for $k = 8$ and $\epsilon = 0.1$. In the figures we report the quality of the solution ((a),(d)), the number of volume computations ((b),(e)) and the number of swaps ((c),(f)) as a function of the number of rows processed so far.

**Dependency on $\epsilon$.** Now we turn our attention to the performance of our algorithm ONLINELS as a function of $\epsilon$. In particular for Shuttle dataset we report how the number of swaps and quality of the solution change as $\epsilon$ changes (experiments on other datasets are available in supplementary material). Interestingly, we can see in Figure 2 that both the number of swaps and quality of the solution decrease smoothly as $\epsilon$ increases. The number of volume computations is very close for all the $\epsilon$.

Finally we note that in our experiment we also consider a variation of our algorithm that does not use a stash $T$. Interestingly we notice that this algorithm has performance very close to ONLINELS.

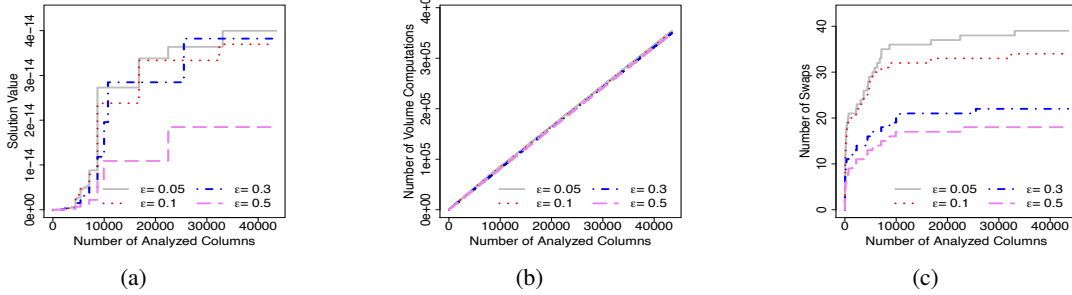

|  (a)  |  (b)  |  (c)  |

Figure 2: Performances of the algorithms on the Shuttle for different values of $\epsilon$ and $k = 8$.

## 5   Conclusion

In this paper, we developed online algorithms for finding the most likelihood configuration of size $k$ for DPPs. The algorithms process data points as they arrive, with small processing time per point and a small total memory footprint. Our main contribution —ONLINE-DPP— achieves a $k^{O(k)}$ multiplicative approximation guarantee with an additive error $\eta$, using memory independent of the size of the data stream.

## Acknowledgments and Disclosure of Funding

Aditya Bhaskara is partially supported by NSF (CCF-2008688) and by a Google Faculty Research Award.

Amin Karbasi is partially supported by NSF (IIS-1845032), ONR (N00014-19-1-2406), AFOSR (FA9550-18-1-0160), and TATA Sons Private Limited.

## Broader Impact

In this paper, we aim to address a fundamental problem in many aspects of data science: how to select a representative and diverse subset of data points. An elegant and intuitive way to score diversity of a subset is through a determinantal point process where diversity is measured via the geometric embedding of the data points. Our paper provides a rigorous and scalable method for maximizing diversity in such probabilistic models.

## Footnotes

[1]The argument needs slightly more care to be made formal; see [Mahabadi et al., 2020] for $\Omega(n)$ space lower bounds for the streaming setting.

[2]By considering the vectors in order and using the base times height formula, the volume can be computed using $O(k^2)$ inner product computations, which is $O(k^2 d)$.

[3]To compute the volume of $k$ columns, Tvol(k), you need to consider the columns in an arbitrary order and find the distance of each column to the span of previous ones. This can be done by maintaining an orthonormal basis of prefixes of this sequence of columns. So for the $i$-th column ($2 \leq i \leq k$), the inner product of it with the previous $i - 1$ columns is needed. So the volume computation is reduced to $O(k^2)$ inner product computations.

[4]Experiments for different $k$ and $\epsilon$ and for the Pen-Based Recognition dataset are similar and postponed to supplementary material

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
