[Supplementary Material]

# Supplementary Material

## A  Online addition with a stash – Full proof

We now present the full proof of Theorem 3.1. We use the following lemmas from [Mahabadi et al., 2019]. We restate them in our notation.

**Lemma A.1.** *Let $P$ be a set of points $Q \subseteq P$ be an approximately (with factor $1+\epsilon$) locally optimum solution for size $k$ volume maximization. In other words, $|Q| = k$, and for any $p \in P \setminus Q$ and $q \in Q$, $vol(Q \cup \{p\} \setminus \{q\}) \leq (1+\epsilon)vol(Q)$. Then $Q$ is an $\alpha$-approximate core-set for $k$-directional height of $P$, where $\alpha$ is $2k(1+\epsilon)$.*

The lemma is powerful because it says that any locally optimum solution is also an $\alpha$-approximate core-set. [Mahabadi et al., 2019] also showed the following.

**Lemma A.2.** *Let $c(P)$ be an $\alpha$-approximate core-set for $k$-directional height for a point set $P$. Then $\mathrm{OPT}_k(P) \leq \alpha^{2k} \cdot \mathrm{OPT}_k(c(P))$.*

Note that the two lemmas imply that any $(1+\epsilon)$-locally optimal solution is a $(2(1+\epsilon)k)^{2k}$ factor approximate solution to the $k$ volume maximization problem. This is because $\mathrm{OPT}_k(Q)$ for a size $k$ set is simply $vol(Q)$, and so Lemma A.2 implies the desired approximation bound.

*Proof of Theorem 3.1.* We will first prove parts 2, 3, 4 of the theorem about memory and running time complexity of Algorithm 1. The size of the stash is clearly upper bounded by the number of times we perform a swap and improve $vol(S)$ by a $1 + \epsilon$ factor (we think about $\epsilon$ as a constant). Therefore its size is at most $\frac{\log \Delta}{\epsilon}$. Since $\epsilon$ is a positive constant, stash $T$ has size $O(\log \Delta)$. This proves the $O(\log \Delta)$ upper bound on the recourse in part 4 of the theorem. So we only keep $k + O(\log \Delta)$ vectors yielding the space complexity bound in part 2.

Every time, a new column $v_t$ arrives, we will check to see if swapping it with any of the $k$ columns in $S$ improves $vol(S)$ by a $1 + \epsilon$ factor. This part consists of $k$ volume computations which is fixed for all steps. Every time set $S$ is updated, we will check the stash for potential further improvements of $vol(S)$. Set $S$ is updated at most $O(\log \Delta)$ times throughout the course of the algorithm. Checking the stash for a swap means computing $vol(S \cup \{v_m\} \setminus \{v_\ell\})$ for every $v_\ell \in S$ and $v_m \in T$. Therefore we may compute the volume of $O(k \log^2 \Delta)$ sets in the while loop of Algorithm 1. This proves both the worst case and amortized upper bound on the running time of each step of the algorithm.

It thus remains to prove the first statement, i.e., the approximation guarantee. We claim that it suffices to prove that $S \cup T$ is an $O(k)$-approximate core-set for $k$-directional height of $\mathcal{V}$. This is because by Lemma A.2, we would then have $\mathrm{OPT}(V) \leq k^{O(k)}\mathrm{OPT}(T \cup S)$. And then, from the description of the algorithm, we have that $S$ is always a $(1 + \epsilon)$ locally optimum solution for $T \cup S$. Using the note following Lemma A.2, we have the desired bound.

Let us show that $S \cup T$ is a core-set for $\mathcal{V}$. We prove this by induction on time $t$. The statement clearly holds until $t = k$. Let $\mathcal{V}_t$ denote $\{v_1, v_2, \dots, v_t\}$. Suppose we had that $S \cup T$ is a $\rho := 2k(1+\epsilon)$ core-set for $k$-directional height for $\mathcal{V}_t$. Now consider $v_{t+1}$. If $v_{t+1}$ was added to $S$, there is nothing to prove (as the point remains forever). Else, we have the guarantee that $S$ is a $(1+\epsilon)$-locally optimal solution for size $k$ volume maximization problem in set $S \cup \{v_{t+1}\}$. Lemma A.1 implies that $S$ is a $\rho$-approximate core-set for $k$-directional height for the set $S \cup \{v_{t+1}\}$, i.e. for any $(k-1)$-subspace $\mathcal{H}$, $\max_{s \in S} d(s, \mathcal{H}) \geq \frac{1}{\rho}d(v_{t+1}, \mathcal{H})$. Together with the inductive hypothesis, we have that $\max_{s \in S \cup T} d(s, \mathcal{H}) \geq \frac{1}{\rho} \max_{v \in \mathcal{V}_{t+1}} d(v, \mathcal{H})$, thus proving the inductive step. Thus $S \cup T$ is a $\rho$-core-set for all $t$. This completes the proof of the theorem. $\qquad\square$

## B  Analysis of ONLINE-DPP – Full proofs

### B.1  Proof of Theorem 3.2

We first show how to prove Theorem 3.2 assuming Theorem 3.4. In what follows, let $S = \cup_\delta S_\delta$.

*Proof of Theorem 3.2.* Let us first bound the space usage. Using (5) and plugging in the value of $\gamma$ from (3), we have a bound of

$$O\left(k\log^2\left((64k)^{2k}\right) + k\log^2\frac{1}{\eta^2}\right) = O\left(k^3\log^2 k + k\log^2\frac{1}{\eta}\right).$$

We then bound the recourse, i.e., the number of times we update the output of the algorithm. We only output $\mathcal{S}$ if its value is at least $\tau$. The volume of a solution never exceeds 1, and every time we update $\mathcal{S}$, its volume doubles. So total number of changes made to the output solution is at most $\log(1/\tau) = O(\log(1/\eta) + k\log k + k\log\log(1/\eta))$.

Now we analyze the running time of each step. For every arriving vector $v_t$ and every $\delta \in \mathcal{D}$, we compute the ridge leverage score, which takes time $T_{\mathsf{ridge}}$. Each $S_\delta$ has size $O(4k\log(8/\gamma)) = O(kL)$ where $L$ is defined in the theorem statement. The number of $\delta$ values in $\mathcal{D}$ is also $\leq L$, and so the running time of each step in Algorithm 2 is the first term of the desired bound. When a $v_t$ is added to some $S_\delta$, we have a new core-set $S$, and therefore we need to rerun the local search algorithm of Mahabadi et al. [2019] on $S$. That local search algorithm starts with the greedy solution as the initialization which takes $k|S|$ volume computations $T_{\mathsf{vol}}(k)$. It then searches for potential improving swaps for at most $O(k\log k)$ iterations as analyzed in Mahabadi et al. [2019]. So the running time of this part is at most $O(k|S| \cdot k\log k \cdot T_{\mathsf{vol}}(k))$. We have an upper bound of $O(kL^2)$ on $|S|$ since it is the union of at most $L$ sets each with size at most $O(kL)$. This proves the desired time bound.

Finally we bound the approximation factor. Suppose the optimal subset of columns of $\mathcal{V}$ (ones with the maximum volume) is $T = \{u_1, u_2, \ldots, u_k\}$. If $\mathrm{vol}(T) \leq \eta$, there is nothing to prove (as we tolerate an additive error $\eta$), and therefore, assume that $\mathrm{vol}(T) > \eta$. We prove in this case, $\mathrm{OPT}(\mathcal{V}) \leq \alpha^k \mathrm{OPT}(S)$, where $\alpha = 8\sqrt{k\log(8/\gamma)}$. Combining this with (4) completes the proof of the approximation bound.

The idea is to use a sequence of swaps, replacing $u_i$ with elements of $S$, as in [Mahabadi et al., 2019]. However, since $S$ is a coreset in the weaker $(\alpha, \gamma)$ sense, we need a slightly more careful argument. Consider the following sequence of swaps, that end up defining sets $T^{(0)}, T^{(1)}, \ldots, T^{(k)}$. Define $T^{(0)} = T$. Next, define $T^{(i)}$ using $T^{(i-1)}$ as follows: if $u_i \in S$, $T^{(i)} = T^{(i-1)}$; otherwise, we remove $u_i$ from $T^{(i-1)}$ and add the vector $w \in S$ that is furthest from $\mathrm{span}(T^{(i-1)} \setminus \{u_i\})$. This defines the set $T^{(i)}$.

To show our theorem we prove using induction that $\mathrm{vol}(T^{(i)}) \geq \frac{\mathrm{vol}(T)}{\alpha^i}$ for all $i \in [k]$. The base case $i = 0$ holds by definition. For any $i$, if we knew that $d(u_i, \mathrm{span}(T^{(i-1)} \setminus \{u_i\})) \geq \gamma$, then we can appeal to the property of an $(\alpha, \gamma)$-core-set to conclude that $\mathrm{vol}(T^{(i)}) \geq \frac{1}{\alpha}\mathrm{vol}(T^{(i-1)})$, completing the inductive step.

So to show our induction step we only need to show that $d(u_i, \mathrm{span}(T^{(i-1)} \setminus \{u_i\})) \geq \gamma$. Now note that by the base times height formula for the volume, we have $\mathrm{vol}(T^{(i-1)})$ to be equal to $d(u_i, \mathrm{span}(T^{(i-1)} \setminus \{u_i\}))$ times the volume of the parallelepiped formed by the vectors in $T^{(i-1)} \setminus \{u_i\}$. As all the vectors have norm $\leq 1$, the latter term is at most 1, and thus

$$d(u_i, \mathrm{span}(T^{(i-1)} \setminus \{u_i\})) \geq \mathrm{vol}(T^{(i-1)}) \geq \frac{\mathrm{vol}(T)}{\alpha^{i-1}} \geq \frac{\eta}{\alpha^{i-1}},$$

where the second last inequality follows form the inductive hypothesis and the last from $\mathrm{vol}(T) \geq \eta$.

Thus, as long as $\eta/\alpha^k \geq \gamma$ we have that $d(u_i, \mathrm{span}(T^{(i-1)} \setminus \{u_i\})) \geq \gamma$ and so we can prove the inductive step. Recalling $L = \log(1/\gamma)$ and writing $E = \log(1/\eta)$, this is equivalent to

$$\gamma \leq \frac{\eta}{(64k\log(8/\gamma))^{k/2}} \iff L \geq E + \frac{k}{2}\log(64k) + \frac{k}{2}\log(3 + L).$$

To ensure this, it suffices to set $L = 2(E + k\log(64k))$. For this choice we have $\frac{k}{2}\log(3 + L) \leq \frac{L}{2}$ (just from the second term). Therefore we have

$$E + \frac{k}{2}\log(64k) + \frac{k}{2}\log(3 + L) \leq \frac{L}{2} + \frac{L}{2} = L.$$

Thus, the choice of $\gamma$ from (3) satisfies the desired properties and concludes the proof that $\mathrm{OPT}(\mathcal{V}) \leq \alpha^k \mathrm{OPT}(S)$. Putting this together with 4 implies that $\mathrm{vol}(\mathcal{S})$ is at least $\mathrm{OPT}(\mathcal{V})/((3k)^{2k} \cdot \alpha^k)$. Since

optimum solution $T$ has volume at least $\eta$ (i.e. $\mathrm{OPT}(\mathcal{V}) \geq \eta$), we deduce that $\mathrm{vol}(\mathcal{S})$ is at least $\tau$. Thus, at any point in the stream where $\mathrm{OPT}(\mathcal{V}) > \eta$, the algorithm will return $\mathcal{S}$ as the solution. This completes the proof of the theorem. $\qquad \square$

### B.1.1 Proof of the packing lemma

The only piece that remains is the proof of Lemma 3.5. It is a geometric statement about vectors in $\mathbb{R}^q$, and the proof uses standard results about determinants of rank-one updates.

Recall the definition of $V_i$ from the statement of the lemma. We define the potential function

$$\phi_i = \det(\delta^2 \mathbf{I}_q + V_i V_i^T).$$

Clearly, we have $\phi_0 = \delta^{2q}$. We first bound $\phi_m$ as follows:

$$\mathrm{tr}(\delta^2 \mathbf{I}_q + V_m V_m^T) = q\delta^2 + \sum_{i \leq m} \|v_i\|^2 \leq q\delta^2 + m,$$

As the trace is the sum of the eigenvalues and the determinant is their product (for a real symmetric matrix), we have, by the AM-GM inequality,

$$\phi_i \leq \left( \frac{q\delta^2 + m}{q} \right)^q \leq \left( \frac{2m}{q} \right)^q.$$

The last inequality follows because $\delta < 1$ and since we can assume that $m > q$ (else there is nothing to prove). Next, the hypothesis of the lemma directly implies (by the matrix determinant lemma) that $\phi_{i+1} > 2\phi_i$. Putting everything together, we have that

$$2^m \delta^{2q} \geq \left( \frac{2m}{q} \right)^q \iff 2^{m/q} \leq \frac{2m}{q} \frac{1}{\delta^2} \iff \frac{2^{m/q}}{m/q} \leq \frac{2}{\delta^2}.$$

We may assume that $m/q \geq 4$ (else we are done). Otherwise since $(m/q) \leq 2^{m/2q}$, the above implies that $2^{m/2q} \leq \frac{2}{\delta^2}$, which implies that $m \leq 4q \log(2/\delta)$.

## B.2 Proof of Lemma 3.6

*Proof.* Both parts of the lemma follow from the following simple claim.

**Claim 3.** *Suppose $M \in \mathbb{R}^{d \times p}$ is a rank $d$ matrix (and so $p \geq d$), and let $v \in \mathbb{R}^d$ be any vector. Then the minimum value of $\|y\|^2$ subject to $My = v$ is precisely $v^T (MM^T)^{-1} v$.*

*Proof of claim 3.* This follows easily from using properties of the pseudo-inverse of $M$, or a direct proof using the SVD of $M$ as follows. Let $L\Sigma R^T$ be the SVD of $M$, where $L, \Sigma \in \mathbb{R}^{d \times d}$, $R \in \mathbb{R}^{p \times d}$, such that $L^T L = R^T R = \mathbf{I}_d$. Now, the $y$ with the minimum norm and $My = v$ is precisely $y = M^\dagger v = R\Sigma^{-1} L^T v$, and thus we have

$$\|y\|^2 = v^T (L\Sigma^{-1} R^T R \Sigma^{-1} L^T) v = v^T (L\Sigma^{-2} L) v = v^T (MM^T)^{-1} v.$$

This completes the proof of the claim. $\qquad \square$

Now consider the matrix $M \in \mathbb{R}^{d \times p}$ that has $p = m + d$ columns. The first $m$ columns are $u_1, u_2, \ldots, u_m$ and the rest are $\delta e_1, \delta e_2, \ldots, \delta e_d$. Clearly, we have $MM^T = \delta^2 \mathbf{I}_d + UU^T$.

To see the first part of the lemma, note that using Claim 3, there exists a $y$ such that $\|y\|^2 \leq c$ and $My = v$. Splitting the coordinates of $y$ into those correspond to $u_i$ and those that correspond to $\delta e_i$ now completes the proof.

The second part can be shown by contradiction. Suppose we can write $v = Ux + z'$, where $\|x\| \leq 1$ and $\|z'\| \leq \delta$. Let $z = z'/\delta$, and consider the vector $y$ formed by the concatenation of $x$ and $z$. Clearly $\|y\|^2 \leq \|x\|^2 + \|z\|^2 \leq 2$. Thus the Claim now gives a contradiction. $\qquad \square$

## B.3 Full proof of Theorem 3.4

We need to show that for any $v_i \in \mathcal{V}$ and any $(k-1)$ dimensional subspace $\mathcal{H}$ of $\mathbb{R}^d$, if $d(v_i, \mathcal{H}) \geq \gamma$, then there exists a $v \in S$ such that $d(v, \mathcal{H}) \geq (1/\alpha) \cdot d(v_i, \mathcal{H})$. Let us fix any such $v_i$ and subspace $\mathcal{H}$. We show that in fact, some element of $S_\delta$ satisfies this property, where $\delta$ is the unique value in $\mathcal{D}$ satisfying $2\delta < d(v_i, \mathcal{H}) \leq 4\delta$. Note that by our choice of parameters, there is always exactly one such $\delta \in \mathcal{D}$. In what follows, let us fix this value of $\delta$, and let $\alpha = \sqrt{64k \log(8/\gamma)}$ be the core-set approximation factor we are aiming to prove in Theorem 3.4. Since all values in $\mathcal{D}$ including $\delta$ are at least $\gamma/2$, we also have $\log(4/\delta) \leq \log(8/\gamma)$. So it suffices to prove that there exists a a $u \in S_\delta$ such that

$$d(u, \mathcal{H})^2 > \frac{\delta^2}{32k \log(4/\delta)}.$$

Now, consider the set $S_\delta$ at the $i$th time step of the algorithm (when we see $v_i$). If $v_i$ gets added to $S_\delta$, there is nothing to prove. Otherwise, either the $M_\delta$ condition was not met, or the size of $S_\delta$ is already at the threshold of $4k \log(4/\delta)$. Let us consider these two cases separately and show the following claims.

**Claim 1.** *For $\delta$ as above, suppose $|S_\delta| < 4k \log(4/\delta)$ at the time that $v_i$ arrives, and suppose that $v_i^T (M_\delta M_\delta^T)^{-1} v_i \leq 2$. Then there exists a $u \in S_\delta$ such that*

$$d(u, \mathcal{H})^2 \geq \frac{\delta^2}{4k \log(4/\delta)}.$$

*Proof of Claim 1.* By using Lemma 3.6, we have that

$$v_i = \sum_{u \in S_\delta} \beta_u u + \delta z, \quad \text{where} \quad \sum_{u \in S_\delta} \beta_u^2 + \|z\|^2 \leq 2. \tag{6}$$

Let $\Pi$ be the projection matrix orthogonal to $\mathcal{H}$. By definition, $d(v_i, \mathcal{H}) \geq 2\delta \iff \|\Pi v_i\| \geq 2\delta$. Thus, squaring and using the Parallelogram law, we have that $2 \left( \left\| \sum_{u \in S_\delta} \beta_u \Pi u \right\|^2 + \delta^2 \|z\|^2 \right) \geq 4\delta^2$. However, by Cauchy-Schwarz, we have that

$$\left\| \sum_{u \in S_\delta} \beta_u \Pi u \right\|^2 \leq \left( \sum_{u \in S_\delta} \beta_u^2 \right) \left( \sum_{u \in S_\delta} \|\Pi u\|^2 \right).$$

Thus if we assume for the sake of contradiction that $\|\Pi u\|^2 < \frac{\delta^2}{|S_\delta|}$ for all $u \in S_\delta$, then we have

$$\left\| \sum_{u \in S_\delta} \beta_u \Pi u \right\|^2 < \left( \sum_{u \in S_\delta} \beta_u^2 \right) \cdot |S_\delta| \frac{\delta^2}{|S_\delta|} \leq \delta^2 \left( \sum_{u \in S_\delta} \beta_u^2 \right).$$

Using (6), we now obtain a contradiction to the inequality we obtained above using the parallelogram law. This completes the proof. $\square$

The more challenging case is when $|S_\delta| \geq 4k \log(4/\delta)$ (and we do not add $v_i$ to $S_\delta$ because of the size threshold). Here we will use the packing lemma to claim that $S_\delta$ has a point with a sufficiently large distance to *any* $(k-1)$ dimensional subspace.

**Claim 2.** *For some $\delta \in (0, 1/2)$, suppose $|S_\delta| = 4k \log(4/\delta)$. Then for* any *$(k-1)$ dimensional subspace $\mathcal{H}$, there exists $u \in S_\delta$ such that $d(u, \mathcal{H})^2 > \frac{\delta^2}{8|S_\delta|}$.*

*Proof of Claim 2.* Suppose for the sake of contradiction that $d(u, \mathcal{H})^2 \leq \frac{\delta}{8|S_\delta|}$ for all $u \in S_\delta$.

Now, let $S_\delta = \{u_1, u_2, \ldots, u_m\}$, where the indices are chosen in the order in which the points were added. Let $P$ be a matrix whose columns are an orthonormal basis for $\mathcal{H}$. Then the projection matrix onto $\mathcal{H}$ is $\Pi_{\mathcal{H}} = PP^T$. Suppose we define new $(k-1)$-dimensional vectors $w_i = P^T u_i$. We show that we have the following properties: (a) $\|w_i\| \leq 1$ for all $i$, and (b) for all $i$,

$$w_i^T \left( \frac{\delta^2}{4} \mathbf{I}_{(k-1)} + W_{i-1} W_{i-1}^T \right)^{-1} w_i > 1. \tag{7}$$

Property (a) is trivial because $\|u_i\| \le 1$ and $P$ is the square root of a projection (and thus has all singular values $\le 1$). To show (b), suppose if possible that (7) does not hold. Then by part 1 of Lemma 3.6 (applied with $\delta/2$ and $c = 1$), we can express

$$w_i = \frac{\delta}{2}z + \sum_{j<i} \beta_j w_j, \quad \text{where} \quad \|z\|^2 + \sum_{j<i} \beta_j^2 \le 1. \tag{8}$$

Now, $Pw_j$ is the projection of $u_j$ to $\mathcal{H}$. Writing $w_j' = u_j - Pw_j$ (the orthogonal component):

$$\left\| u_i - \sum_{j<i} \beta_j u_j \right\|^2 = \left\| \frac{\delta}{2}Pz + w_i' - \sum_{j<i} \beta_j w_j' \right\|^2$$

$$\le 2 \left[ \frac{\delta^2}{4} \|Pz\|^2 + \left\| w_i' - \sum_{j<i} \beta_j w_j' \right\|^2 \right]$$

$$\le 2 \left[ \frac{\delta^2}{4} + |S_\delta|(1 + \sum_{j<i} \beta_j^2) \cdot \max_j \|w_j'\|^2 \right] \tag{9}$$

To get to (9) we have used the Cauchy-Schwarz inequality, along with the bound $\|Pz\| \le 1$, which follows from (8). Now, plugging in the assumption that $\|w_j'\|^2 \le \delta^2/(8|S_\delta|)$ (i.e., the $u_i$'s all have a small component orthogonal to $\mathcal{H}$), and using the simple bound of $(1 + \sum_{j<i} \beta_j^2) \le 2$ (which follows from (8)), we have that

$$\left\| u_i - \sum_{j<i} \beta_j u_j \right\|^2 \le \delta^2.$$

In other words, $u_i$ can be expressed as $z' + \sum_{j<i} \beta_j u_j$, where $\|z'\| \le 1$ and $\sum_j \beta_j^2 \le 1$. By the second part of Lemma 3.6, this now implies that at the time $u_i$ was being added, we have

$$u_i^T (M_\delta M_\delta^T)^{-1} u_i \le 2,$$

which is a contradiction because in this case, we would not have added $u_i$ to $S_\delta$. Thus, we conclude that (b) holds.

Finally, we apply the packing lemma (Lemma 3.5) to the vectors $\{w_i\}_{i=1}^m$ with $\delta/2$ and $q = (k-1)$, which gives us that $m \le 4(k-1)\log(4/\delta) < 4k\log(4/\delta)$, a contradiction. This completes the proof of the claim. □

The claim immediately implies that if $|S_\delta|$ was $4k\log(2/\delta)$, it does not matter that $u_i$ was not added to $S_\delta$. For any $\mathcal{H}$, we already have $\max_{u \in S_\delta} d(u, \mathcal{H})$ being $\ge (1/\alpha) \cdot 4\delta \ge (1/\alpha) \cdot d(u_i, \mathcal{H})$. This completes the proof of the theorem.

## C   Removing the boundedness assumption on vector lengths

We now show how to remove the assumption that $\|v_t\| \le 1$ for all $v_t \in \mathcal{V}$. Thus in this case, all we are given is an additive error parameter $\eta$, and an upper bound $B$ on the maximum norm. I.e., $B$ satisfies $B \ge \max_t \|v_t\|$.

In fact, as we will see, our bounds only depend on $M = \lceil \log B \rceil$. Thus, having an upper bound on $M$ up to a constant suffices (and this is much weaker than having a good bound on $B$).

The modification to the algorithm is now straightforward: once a vector $v_t$ arrives, we scale it down by a factor $B$, and we use $\eta' = \frac{\eta}{B^k}$ in place of $\eta$ in the algorithm (i.e., in the definitions of $\gamma$, $\mathcal{D}$, etc.). Note that scaling each vector by $B$ changes the volume of every $k$-simplex by exactly $B^k$, so if vol$'$ denotes the volume after scaling, then vol$'(S) = \frac{1}{B^k}$vol$(S)$. Thus, if we had vol$'(S) \ge \beta \cdot$ vol$'(S^*) - \eta'$, then vol$(S) \ge \beta \cdot$ vol$(S^*) - \eta$.

Plugging this into Theorem 3.2, we end up with a multiplicative approximation factor of $((k + k\log B + \log(1/\eta))^{O(k)}$ and a space complexity of $O\left(k^3 \log^2 kB + k\log^2 \frac{1}{\eta}\right)$ vectors.

# D  Additional experiments

In this section, we present additional experiments. In particular, we present the results the Pen-Based Recognition dataset and we present the missing experiments on the dependency on $\epsilon$. Finally we show how the performance of our algorithm change as we change $k$ but we keep $\epsilon$ fix to $\epsilon = 0.1$ for the Spambase dataset and the Statlog dataset and $\epsilon = 0.05$ for the Pen-Based Recognition dataset.

## D.1  Quality, consistency and amount of computation for the Pen-Based Recognition dataset

In Figure 3 we show the quality, the number of volume computations and the consistency for the Pen-Based Recognition dataset. Also in this case we see that the final solution of ONLINEGREEDY is comparable with the solution of offline GREEDY, the solution of ONLINELS is slightly worse than it. Furthermore ONLINELS and ONLINEGREEDY make a similar number of computation but that ONLINELS is significant more consistent.

(a)  (b)  (c)

Figure 3: Performances of the algorithms on the Pen-Based Recognition for $k = 8$ and $\epsilon = 0.05$. In the figures we report the quality of the solution((a)), the number of volume computations((b)) and the number of swaps((c)) as a function of the number of rows processed so far.

## D.2  Dependency on $\epsilon$ for the Pen-Based Recognition dataset and the Spambase dataset

In Figure 4 we present how the performance of ONLINELS as a function of $\epsilon$. Also in this case we note a smooth trade-off between quality and efficiency.

(a)  (b)  (c)

(d)  (e)  (f)

Figure 4:  Performances of the algorithms for the Pen-Based Recognition((a),(b),(c)) dataset((a),(b),(c)) and the Spambase dataset((d),(e),(f)) for different values of $\epsilon$ and $k = 8$.

### D.3 Experiments for different values of $k$

In this subsection we present experiments for different values of $k$ for the three datasets. In the experiments we use $\epsilon = 0.1$ for the Spambase dataset(Figure 6) and the Statlog dataset(Figure 7) and $\epsilon = 0.05$ for the Pen-Based Recognition dataset(Figure 5). We experiment for $k = 4$ and 16(for Statlog dataset we only experiment with $k = 4$ because the dataset has 9 dimensions). Overall the results are consistent with the results for $k = 8$. ONLINELS has similar performance to ONLINEGREEDY but it is significantly more consistent and the final solution of both online algorithm is close to the offline solution of ONLINEGREEDY. Although we note that as $k$ grow the gap between the online and offline solution grows as well.

(a) $k = 4$      (b) $k = 4$      (c) $k = 4$

(d) $k = 16$      (e) $k = 16$      (f) $k = 16$

Figure 5: Performances of the algorithms on the Pen-Based Recognition dataset for $\epsilon = 0.05$. In the figures we report the quality of the solution((a),(d)), the number of volume computations((b),(e)) and the number of swaps((c),(f)) as a function of the number of rows processed so far.

## E  Necessity of the stash

The following example shows that a local search algorithm that online keeps only the best solution without a stash cannot achieve any bounded approximation guarantee. In particular this implies that the algorithm ONLINEGREEDY (which performs quite well in experiments) has unbounded worst case guarantees.

We give a simple example with $k = 2$. Let $r > 2$ be an integer and consider the vectors:

$$v_0 = (1, 0, y)$$
$$v_1 = (1, x, -y)$$
$$v_2 = (1, 2x, y)$$
$$\vdots$$
$$v_r = (1, rx, (-1)^r y)$$

The optimal solution is to choose the first and the last vectors. It turns out that the squared area of the parallelogram spanned is $r^2 x^2 y^2 + r^2 x^2$.

Now consider the execution of the ONLINEGREEDY algorithm. It starts by choosing $v_0$ and $v_1$ as the solution. We claim that after seeing $v_2$, it drops $v_0$ and maintains $v_1, v_2$ as the solution. More generally, we claim that the algorithm always maintains the last two vectors it has seen as the solution. We show this by induction. Consider the $(t+1)$th time step. The inductive hypothesis implies that the

Figure 6: Performances of the algorithms on the Spambase dataset for $\epsilon = 0.1$. In the figures we report the quality of the solution((a),(d)), the number of volume computations((b),(e)) and the number of swaps((c),(f)) as a function of the number of rows processed so far.

Figure 7: Performances of the algorithms on the Statloog dataset for $\epsilon = 0.1$. In the figures we report the quality of the solution((a),(d)), the number of volume computations((b),(e)) and the number of swaps((c),(f)) as a function of the number of rows processed so far.

solution before seeing $v_{t+1}$ is $\{v_{t-1}, v_t\}$. Now consider the vectors $v_{t-1}, v_t, v_{t+1}$. A straightforward calculation now shows the following:

$$\text{vol}^2(v_{t-1}, v_t) = (2t-1)^2 x^2 y^2 + x^2 + 4y^2$$
$$\text{vol}^2(v_{t-1}, v_{t+1}) = 4x^2 y^2 + 4x^2$$
$$\text{vol}^2(v_t, v_{t+1}) = (2t+1)^2 x^2 y^2 + x^2 + 4y^2.$$

Now, if we choose $x = y$ and both small enough, $\text{vol}(v_t, v_{t+1})$ has the largest volume. Because of this, the algorithm ONLINEGREEDY will always keep only the last two vectors in its solution.

Thus, at the end of the stream, $v_r$, the ratio between the volume that the algorithm ends up with and the optimal is:

$$\frac{r^2 x^2 y^2 + r^2 x^2}{(2r-1)^2 x^2 y^2 + x^2 + 4y^2} \approx r,$$

assuming $x, y \approx 1/\sqrt{r}$. Thus the approximation ratio of ONLINEGREEDY, even with $k = 2$, grows linearly with the number of vectors in the stream even in this simple three-dimensional example.