[Reviews · NeurIPS 2020]

Review 1

Summary and Contributions: Discrete Determinantal Point Processes (DPPs) are probability distributions over subsets of a finite set of (feature) vectors where the probability of each subset is proportional to the volume of the parallelogram that they span. Since these distributions capture the negative correlations between items, they have been extensively employed in subset selection tasks where diversity is preferred. MAP estimation for DPPs is a well-studied problem which formally given a set of vectors and an integer $k$ asks a for a subset of $k$ items with (approximately) highest volume (probability); this objective functions can be translated as the most diverse subset of size $k$. The problem is shown to be hard. In particular, there is an exponential $O(1)^k$ lower bound on the approximation factor in the offline setting. This work studies the MAP estimation for DPPs in a specific streaming setting as follows: Vectors arrive in a stream and upon arrival of any vectors, the algorithm has to decide either store it (permanently) in the memory or erase it. Selected vectors can not be deleted from the memory after this step. Moreover, the number of vectors in the stream is not known in advance. MAP estimation has been previously studied in a different distributed setting known as core-sets. An application of that workstream yields an algorithm in the above setting with approximation factor $O(k)^(k/\epsilon)$ with a peak memory of $O(n^\epsilon * kd)$ for $n$ being the number of points and $d$ the ambient dimension. The main contribution of this work is to remove this dependency on $n$ in the space, while roughly maintaining the approximation guarantee. They also show that although their algorithm is able to maintain the guaranteed approximation factor at any point in the stream, the total number of changes in the solution is bounded by poly(k). They do it in two steps: 1. Built upon the work of [1], they propose a local search method with approximation factor $k^O(k)$, however, the space complexity depends on the logarithm of the ratio of the optimal volume to the volume of the first $k$ points in the stream. 2. They offer an improved algorithm that given a parameter $r$, achieves an $O(k + log(1/r))^O(k)$ approximation factor (with an additional additive error of $r$) with memory $poly(k, log(1/r))$. 1. Sepideh Mahabadi, Ilya Razenshteyn, David P Woodruff, and Samson Zhou. Non-adaptive adaptive sampling on turnstile streams. arXiv preprint arXiv:2004.10969, 2020.

Strengths: DPPs are extensively used for modeling subset selections in the real-world tasks, with real-time and huge data sets. In this regime, streaming algorithms for inference from DPPs seems very critical. Their algorithm in terms of memory and time complexity seems very practical and appealing in this setting. The guaranteed approximation factor does also seems reasonable given the hardness results in various settings of the problem. The proofs are also elegant and precise.

Weaknesses: I don't see any significant weakness to include here.

Correctness: Yes, I confirm the theoretical correctness of the results and soundness of experiments.

Clarity: Yes.

Relation to Prior Work: Yes.

Reproducibility: Yes

Additional Feedback: - In subsection 3.1, given the algorithm the following statement does not seem precise: "Let vol_first(V) be the volume of the parallellepiped formed by the first k vectors of the stream"; It will be the first parallelopiped with non zero volume that can be formed in the stream. - In some applications of DPP, the actual features vectors are not available, e.g. when vectors are passed through another kernel line an RBF kernel. Does your algorithm can be adapted in those settings as well? Let's say at each step a single row of the PSD kernel of the DPP is revealed. - In the last paragraph of the intro the algorithm of [Derezinski et al., 2019] is referred as an MCMC algorithm which is not the case. That is a spectral algorithm. - Out of curiosity I was thinking if your analysis in the first part is tight and I came up with this example. d=k and we have 2k vectors in the stream. First k vectors are standard vectors and the second k vectors are rows of a Hadamard matrix ( k dimensional +1, -1 vectors which are mutually orthogonal). your algorithm does not select any of the second k vectors and the approximation factor would be k^(k) in this case.


Review 2

Summary and Contributions: This paper aims to design an online algorithm for MAP inference of Determinantal Point Processes (DPPs). The authors observe that a naive online greedy local search algorithm (i.e., keep the stream input if replacement with it yields the improvement) provides an unbounded approximation ratio. And they propose a stash-based algorithm, i.e., if an item is swapped then it is stored to a stash and used for future swap operations, and prove the k^O(k) approximation ratio for budget k. Furthermore, to drop off the conditional number dependency, they devise a coreset based algorithm that introduces an additive error term. Experimental results demonstrate the proposed algorithm is comparable to the online greedy but less number of swaps.

Strengths: This is the first work on the online DPP MAP inference with provable approximation guarantees. Unlike other prior works, this result does not require a condition on the minimum eigenvalue for non-negativity and provides a multiplicative guarantee. The authors also give a simple counter-example that the naive online greedy algorithm fails, i.e., can return arbitrarily small approximation guarantee.

Weaknesses: Although the paper provides a coreset based algorithm (Online-DPP) independent on the conditional number of volumes, there is a lack of empirical results of this. It would be much interesting if empirical evidence that Online-DPP requires a smaller running time than Online-LS with a small vol_first is provided.

Correctness: The proof techniques in this work are novel and seem to be correct.

Clarity: This paper is fairly well-written and easy to comprehend. The analysis and proofs are given with clear intuitions and straightforward. But there are some minor typos: - Line numbers in section 2 (at line 116) are missing. - In line 148, does vol_{min} imply vol_{first}? - Does Shuttle dataset refer to Statlog?

Relation to Prior Work: This is the first work that demonstrates online DPP MAP inference with theoretical analysis.

Reproducibility: Yes

Additional Feedback:


Review 3

Summary and Contributions: The paper proposes two algorithms for the online MAP inference of DPP. The first one achieves the best-known approximation ratio while the space complexity can be arbitrarily bad due to the (logarithmic) dependence on a condition number. The second one overcomes the shortcoming at a slight sacrifice of the approximation guarantee; consequently, its space complexity depends only on a polynomial of k, the size of output, and a controllable parameter. With both algorithms, the number of changes to solutions is also bounded. Experiments compare the first algorithm with two baselines. -- after rebuttal -- Thank you for kindly responding to my comments. I read the rebuttal and I will maintain my score.

Strengths: - The paper addresses an important problem setting and makes strong theoretical results. - Theoretical advantages over prior work are clearly described. - The results are explained clearly.

Weaknesses: - Experiments do not seem to be convincing since (1) only the first algorithm (ONLINE-LS) is evaluated, (2) only simple baselines are considered, and (3) the improvement in the number of swaps do not seem to be so significant in practice.

Correctness: The claims seem to be correct, although I did not check the details of the proofs.

Clarity: The paper is well written and the results are clearly described.

Relation to Prior Work: Relation to previous work is mostly well described, but the authors should mention recent advances in submodular maximization. For example, [Bian et al. (2017), Guarantees for Greedy Maximization of Non-submodular Functions with Applications] studies maximization of determinantal functions with approximate submodularity, and [Harshaw et al. (2019), Submodular Maximization beyond Non-negativity: Guarantees, Fast Algorithms, and Applications] studies maximization of submodular functions whose value can be negative.

Reproducibility: No

Additional Feedback:


Review 4

Summary and Contributions: The authors tackle the problem of finding a k-subset of vectors maximizing the corresponding squared volume in the online setting, building on the work of Mahabadi et al. [2020] regarding the streaming setting. The simple swapping and leverage score based algorithms considered are shown to have poly-log(k) memory footprint and output a subset whose square volume is at most k^O(k) far from the optimal solution.

Strengths: The memory footprint of the proposed algorithms is independent of the length of the stream and scale as poly-log(k). Some linear algebraic Lemmas might be of independent interest to the community.

Weaknesses: The proposed methodology works for small values of k, only. It is not clear whether the output of Algorithm 2 (Online-DPP) has size k, while the is the very purpose of the paper. There is no comparison with streaming algorithms, nor with a random baseline such as the best k-subset among a fixed number of uniformly random draws. For this kind of applied contribution, supplementary code might be expected, please consider releasing the code upon acceptance.

Correctness: The title must be clearly state that the method applies to k-DPPs and not DPPs. It is not clear at all whether the output of Algorithm 2 Online-DPP has size k, while this is the purpose of the paper as presented in the Preliminary section. In the experimental section (Section 4) the magnitude of the solution values displayed on the plots seem to be of the order of machine precision ~10^-14, isn't it problematic ? How useful is the exponential k^O(k) approximation factor of the procedure claimed in Theorems 3.1-2 ?

Clarity: The paper is mostly well written. Some inconsistencies are listed in "Additional feedback".

Relation to Prior Work: The goal of the paper seems to adapt the work of Mahabadi et al., [2020] made in the streaming setting to the very close online setting.

Reproducibility: Yes

Additional Feedback: The title must be clearly state that the method applies to k-DPPs and not DPPs. Is there a reason to favor the determinantal point process terminology instead of D-experimental design? Please give the time complexity of Tridge(kL) + Tvol(k). l 108 Derezinski & al generate exact DPP samples and not approximate samples via MCMC l 116' problem 2 is not defined -> (2) There are some inconsistencies in the notation: - citet / citep, e.g., l18 49 95 99 164 - vol / it{vol} - vol^2 / vol()^2 - some matrices are noted with plain math (V), others with mathbb (L), others with mathbf I - some sets have mathcal notation, others don't Suggestions to improve readability - make sure hyperlinks are clickable in the main paper - unnecessary brackets, e.g. legend of Figure 1, l62-75. - adapt the vol notation to remove the OPT notation which is confusing - instead of using "in supplementary, section xx" I suggest "in Appendix xx"

[Author Response · NeurIPS 2020]

We thank all the reviewers for their work in this challenging time. We will fix all typos and apply the clarifications
suggested in the reviews. Below, we address specific questions and concerns.

**Reviewer 1.** **Q1:** *In some applications of DPP, the actual features vectors are not available,[...]*
**A1:** Thanks for suggesting the problem. While the Online-DPP algorithm only requires computing ridge leverage
scores, it is not clear if the entire analysis goes through. This is an interesting point that we will verify.

**Q2:** *Out of curiosity I was thinking if your analysis in the first part is tight and I came up with this example.*
**A2:** Beautiful! Thanks for this observation :)

**Reviewer 2.** **Q1:** *Experimental focus on Online-LS and not Online-DPP*
**A1:** Thanks for the suggestion, we decided to focus our experiments on Online-LS because it is a more practical
approach to the problem. In line 148, $vol_{\min}$ should be $vol_{\text{first}}$. Finally, Shuttle dataset refers to Statlog, we will clarify
this in the final version.

**Reviewer 3.** **Q1:** *Experiments do not seem to be convincing [...]*
**A1:** Thanks for raising those points. For (1), please refer to the answer to Reviewer 2. For (2), we compared with a
natural heuristic for the problem, it is not clear to us how to define an alternative heuristic for the online problem. An
additional advantage of Online-Greedy is that it is a simplified version of Online-LS and so it shows how our theoretical
results compare with experiments. For (3), we think that it is interesting that Online-LS (a) always seems to do strictly
better, (b) the gap is not too large on chosen datasets even though we show that it can be arbitrarily bad (Appendix E). It
suggests that the choice of algorithm in practice depends on how much accuracy we care about.

We will also add references to the recent works on submodular optimization.

**Reviewer 4.** **Q1:** *The proposed methodology works for small values of $k$ only.*
**A1:** Our algorithm works for all $k$. The approximation factor depends exponentially on $k$ but this is unavoidable for
this problem *even in the offline setting* unless P=NP (see answer to Q6)

**Q2:** *It is not clear whether the output of Algorithm 2 (Online-DPP) has size $k$, while the is the very purpose of the paper*
**A2:** Algorithm 2 (Online-DPP) provides a *core-set* (a set that is guaranteed to contain a $k$-subset that provides a good
approximation). As is standard, a core-set has size larger than $k$. In lines 199-209, we explain how to maintain a
solution of size exactly $k$ over the course of the algorithm.

**Q3:** *There is no comparison with streaming algorithms, nor with a random baseline[...]*
**A3:** Streaming algorithms do not optimize the number of times the maintained solution "changes" (which is the key
parameter in the online version). Our Online-Greedy and Online-LS may be viewed as variants of streaming algorithms
where we do not change the solution unless the value improves considerably. Moreover, we show that our solution
quality is comparable with the *offline* algorithm, which is much stronger than the streaming and random baselines.

**Q4:** *Supplementary code, and proposed change to the title.*
**A4:** We will make the code publicly available once the paper is accepted, and will consider the title change, thanks!

**Q5:** *Magnitude of the solution in the experiments approaches machine precision?*
**A5:** In our experiment (note that we track the change of the cost of the solution after each column insertion) we did not
observe any sort of numerical instability.

**Q6:** *How useful is the exponential $k^{O(k)}$ approximation factor [...]*
**A6:** An exponential dependence on $k$ in the approximation factor is the best possible guarantee unless $P = NP$ (as
discussed in lines 39-40), even in the *offline* setting. This is effectively because volume in $k$ dimensions scales as the
$k$th power. This is why some works use $vol()^{1/k}$ as the measure of interest.

**Q7:** *Please give the time complexity of Tridge(kL) + Tvol(k).*
**A7:** The time complexity of Tridge(kL) is discussed in lines 185-187. To compute the volume of $k$ columns, Tvol(k),
you need to consider the columns in an arbitrary order and find the distance of each column to the span of previous
ones. This can be done by maintaining an orthonormal basis of prefixes of this sequence of columns. So for the $i$-th
column ($2 \leq i \leq k$), the inner product of it with the previous $i - 1$ columns is needed. So the volume computation is
reduced to $O(k^2)$ inner product computations. We will clarify this in the final version of the paper.

We will incorporate the other proposed suggestions (including fixing the reference to Derezinski et al.) and typo fixes to
improve the readability. The *DPP terminology* (instead of optimal design) is preferred mainly to be consistent with
recent work on coresets (which is why we mention optimal designs in related work).

[Meta-Review · NeurIPS 2020]

The paper proposes an algorithm for Online-DPPs, which has interesting approximation guarantees. While the work is an extension of Mahabadi et al, 2020, the required extensions are non-trivial. The paper shows improvements over a greedy baseline and is theoretically well-founded. Weaknesses of the paper include the experimental evaluation, where improvement over other methods was present but not very clear. The recommendation for NeurIPS is accept.